# Fingerprinting, Antimicrobial, Antioxidant, Anticancer, Cyclooxygenase and Metabolic Enzymes Inhibitory Characteristic Evaluations of *Stachys viticina* Boiss. Essential Oil

**DOI:** 10.3390/molecules24213880

**Published:** 2019-10-28

**Authors:** Nidal Jaradat, Nawaf Al-Maharik

**Affiliations:** 1Department of Pharmacy, Faculty of Medicine and Health Sciences, An-Najah National University, Nablus, 00970, Palestine; 2Department of Chemistry, Faculty of Science, An-Najah National University, Nablus, 00970, Palestine

**Keywords:** *Stachys viticina*, essential oil, cytotoxicity, antioxidant, antimicrobial, metabolic enzymes, cyclooxygenase

## Abstract

The present study aimed to identify the chemical constituents and to assess the in-vitro, antimicrobial, anticancer, antioxidant, metabolic enzymes and cyclooxygenase (COX) inhibitory properties of essential oil (EO) of *Stachys viticina* Boiss. leaves. The *S. viticina* EO was isolated and identified using microwave-ultrasonic and GC-MS techniques, respectively. Fifty-two compounds were identified, of which *endo*-borneol was the major component, followed by eucalyptol and epizonarene. The EO was evaluated against a panel of in-vitro bioassays. The EO displayed antimicrobial activity against methicillin-resistant *Staphylococcus aureus* (MRSA), *Escherichia coli* and *Epidermophyton floccosum*, with MIC values of 0.039, 0.078 and 0.78 mg/mL, respectively. The EO exhibited cytotoxicity against HeLa (cervical adenocarcinoma) and Colo-205 (colon) cancer cell lines with percentages of inhibition of 95% and 90%, for EO concentrations of 1.25 and 0.5 mg/mL, respectively. Furthermore, it showed metabolic enzyme (α-amylase, α-glucosidase, and lipase) inhibitory (IC_50_ = 45.22 ± 1.1, 63.09 ± 0.26, 501.18 ± 0.38 µg/mL, respectively) and antioxidant activity, with an IC_50_ value of 19.95 ± 2.08 µg/mL. Moreover, the *S. viticina* EO showed high cyclooxygenase inhibitory activity against COX-1 and COX-2 with IC_50_ values of 0.25 and 0.5 µg/mL, respectively, similar to those of the positive control (the NSAID etodolac). Outcomes amassed from this investigation illustrate that *S. viticina* EO represents a rich source of pharmacologically active molecules which can be further validated and explored clinically for its therapeutic potential and for the development and design of new natural therapeutic preparations.

## 1. Introduction

People since ancient times have been using plant secondary metabolites in their daily life for the treatment of a wide range of diseases, as food preservatives and flavoring, and to annihilate insects. The World Health Organization has estimated that around 80% of the world population rely on plant extracts for their medication [1]. With time, synthetic chemicals have replaced plant secondary metabolites as the former provide early results. However, most synthetic drugs have various side effects and may cause many serious human health problems, therefore strategies have been implemented to replace synthetic drugs with plant secondary metabolites due to their ability to protect human body against oxidative damage and their great positive impact on human health [1,2,3]. Nowadays, due to the growing public awareness about the probable harmful effects of synthetic additives and the tendency of consumers to use natural foods, the use of natural bioactive compounds is witnessing significant growth [4]. Due to the wide prevalence of cancer worldwide, there is an essential and urgent need for a search for new anticancer medications [5]. In fact, majorities of used antitumor drugs are derived from natural sources [6].

The essential oils extracted from plants have been used in traditional medicine for the treatment and prevention of various diseases, in cosmetics, foods, and food supplements in addition to dental practice and hygiene products [7,8].

*Stachys* (Lamiaceae family) is a large genus of herbs and shrubs containing around 300 species, widely spread in the temperate regions, especially in the Irano-Turanian and Mediterranean areas. Recently, several studies have documented anti-inflammatory, cytotoxic, antioxidant, immune system booster and antimicrobial properties of the extracts from a number of *Stachys* species [9].

*Stachys viticina* Boiss. is a perennial herbaceous shrubby plant that can reach a height of 100 cm, with multiple branches, grey-colored and erect stems. The upper leaves are crenate, while the lower ones are large, oblong-ovate and verticillaster. In traditional medicine, various *Stachys* species have been utilized as anti-inflammatory, antimicrobial, anti-diarrheal, wound healing and astringent remedies. In addition, they were used to treat ulcers, cough, sclerosis of the spleen and genital tumors [10,11].

It is worth mentioning that the chemical constituents of *S. viticina* EO, as well as its biological and pharmacological properties, haven’t been reported. Thus, the current study was designed to investigate the chemical composition of the EO extracted from *S. viticina* leaves. Additionally, the antioxidant, anticancer, antimicrobial, metabolic enzyme (*α*-amylase, *α*-glucosidase, and lipase) and cyclooxygenase (COX) inhibitory properties of *S. viticina* EO was assessed.

## 2. Results

### 2.1. Phytochemical Composition of S. viticina Essential Oil

In accord with our ability and experience in the isolation and identification of natural products from plants, as well as in assessing their biological activities, we decided to study the chemical components of *S. viticina* EO, as well as their biological effects, in the hope to finding new drug leads from natural sources. The isolation procedure was carried out using a microwave-ultrasonic apparatus. The phytochemical composition was identified and estimated using GC-MS. The obtained yield of the EO was 1.72 ± 0.97%. GC-MS analyses indicated the presence of 52 phytochemical compounds in *S. viticina* EO, of which *endo*-borneol was the major component, followed by eucalyptol and epizonarene, as revealed in Table 1 and Appendix A.

### 2.2. Antioxidant and Metabolic Enzymes Inhibitory Activity

The inhibitory activity of *S. viticina* EO against oxidation and metabolic enzymes (α-amylase and α-glucosidase and lipase) was assessed using standard biomedical assays. The results showed that *S. viticina* essential oil has α-amylase and α-glucosidase inhibitory activities, with IC_50_ values of 45.22 ± 1.1 and 63.09 ± 0.26 µg/mL, respectively.

As indicated in Table 2 and Appendix A, the EO displayed antioxidant and antilipase activities with IC_50_ values of 19.95 ± 2.08 and 501.18 ± 0.38 µg/mL, respectively.

### 2.3. Antimicrobial Capacity

The microdilution assay was used to determine the antimicrobial potential of *S. viticina* essential oil against Methicillin-resistant *Staphylococcus aureus* (MRSA), *S. aureus*, *E. faecium*, *S. sonnie*, *P. aeruginosa*, *E. coli*, *E. floccosum*, and *C. albicans*. The EO inhibited the growth of all of the studied bacterial and fungal strains to different degrees. The highest antibacterial activity was recorded against MRSA, followed by *E. coli* with MIC values of 0.039 and 0.078 mg/mL, respectively, while the highest antifungal activity observed was against *E. floccosum* with MIC value of 0.78 mg/mL as presented in Table 3.

### 2.4. Cyclooxygenase Inhibitory Effect

The cyclooxygenase enzymatic inhibitory activity assay of *S. viticina* EO was conducted utilizing Cayman Chemical ELISA kit No. 560131. The calculated percentage inhibitions of COX-2 as well as for COX-1 were 98% and 99% at EO concentrations of 0.25 and 0.5 µg/mL, respectively. The results showed that the *S. viticina* EO displayed a strong, but not selective inhibition activity towards both COX-2 and COX-1 enzymes.

### 2.5. Cytotoxic Effect

As shown in Figure 1, treatment of Colo-205 cells with 1, 0.5 and 0.25 mg/mL of EO induced cytotoxicity significantly (*p* ≤ 0.0001) by approximately 90%, 90%, and 55%, respectively. Treatment of HeLa cells with 2.5, 1.25 and 0.625 mg/mL of EO induced cytotoxicity significantly (*p* ≤ 0.0001) by approximately 95%, 95%, and 80%, respectively, while at a concentration of 0.1875 mg/mL did not exhibit a significant effect as presented in Figure 2.

## 3. Discussion

The study of plants secondary metabolites represents a worldwide strategy for the search of new therapeutic agents since plants contain a wide range of biologically active compounds. Medicinal plants are extensively used in traditional medicine due to their healing properties. In order to set up an inventory of their biological activities, it is important to link scientific studies with clinical observations and traditional knowledge. This investigation is part of this scenario and was aimed to screen the biological activity of *Stachys viticina* essential oil. To the best of the authors’ knowledge, the full chemical constituents, as well as the antimicrobial, anticancer, antioxidant, metabolic enzymes and cyclooxygenase inhibitory properties of *S. viticina* EO have not been reported to date.

### 3.1. Phytochemical Components of S. viticina Essential Oil

The *S. viticina* EO was isolated using an eco-friendly microwave-ultrasonic technique and the components of the EO were identified using GC-MS technology. The primary components were determined to be *endo*-borneol, eucalyptol, epizonarene, α-cadinene, bornyl chloride, and benzylin with percentage areas of 29.06 ± 1.24, 21.26 ± 1.97, 7.89 ± 0.91, 6.09 ± 0.1, 5.79 ± 0.09, and 5.29 ± 0.31, respectively. The chromatogram of *S. viticina* EO is shown in Appendix A. The retention times and mass fragmentation data of the 52 compounds identified are listed in Table 1.

Recently, Gören et al. reported the identification of only 20 compounds from EO chemical composition of *S. viticina* aerial parts (flowering tops, stems, and leaves) growing in Turkey, of which *β*-caryophyllene (62.3%), farnesyl acetate (8.9%) and *α*-bisabolol (4.4%) were the major components [12]. The difference in the results could be attributed to the difference in the area of plant growth, extraction conditions and/or to the fact that only the leaves were used in this investigation, while in Goren et al. study the aerial parts (flowers, leaves and stems) were used.

### 3.2. Metabolic Enzymes Inhibitory Activity

The metabolic pancreatic *α*-amylase and the intestinal *α*-glucosidase enzymes play a vital role in the digestion of carbohydrates such as starch, dextrin, maltotriose, maltose, glucose and table sugar. In fact, diabetes, overweight and obesity can be controlled by the inhibition of these enzymes due to their ability to retard the carbohydrate’s digestion process and to improve glucose tolerance in patients suffering from type II of diabetes [13].

As shown in Appendix A, *S*. *viticina* EO displayed *α*-amylase pancreatic enzyme and *α*-glucosidase inhibitory activity in vitro, with IC_50_ values of 45.22 ± 1.1 µg/mL and 63.09 ± 0.26 µg/mL, respectively, compared to that of the antidiabetic drug acarbose which has *α*-amylase and *α*-glucosidase inhibitory effects with IC_50_ values of 28.89 ± 1.22 µg/mL and 37.15 ± 0.33 µg/mL, respectively. This means that *S. viticina* EO inhibited 56.52% and 69.82% of *α*-amylase and *α*-glucosidase activity, respectively.

There are many reports on the inhibitory effect of EO extracted from different plant species on the α-amylase and α-glycosidase. Loizzo et al. reported that the *Cedrus libani* wood EO exhibited an *α*-amylase inhibitory effect with IC_50_ of 140 µg/mL [14]. Ma et al. found that the EOs extracted from *Pyrola calliantha* entire plant, *Senecio scandens* flowers and of *Schisandra chinensis* roots have α-glycosidase inhibitory activity with IC_50_ values of 22.11, 130.4 and 19.25 µg/mL, respectively [15]. Another study conducted by Dang et al. revealed that the *Citrus medica* var. *sarcodactylis* (Buddha’s hand) fruits EO inhibited an *α*-glucosidase enzyme with an IC_50_ value of 412.2 μg/mL [16].

However, the IC_50_ values showed that *S. viticina* EO displayed a stronger inhibitory activity on α-amylase than on α-glucosidase activity, and therefore this finding is therapeutically important especially in preventing some of the side effects associated with the use of chemical α-glucosidase and α-amylase inhibitory agents [17]. The observed α-amylase and α-glucosidase inhibitory activity of the *S*. *viticina* EO gives credence to the fact that *S. viticina* EO could be promising antidiabetic medicine after in vivo and clinical trials.

Therapeutic agents that can inhibit the pancreatic lipase enzyme have gained much importance in recent years. Obviously, the inhibition of pancreatic lipase enzyme retards the lipids digestion and consequently lower the rate of fats absorption and decrease the levels of triglyceride in blood serum which may also cause a reduction of body weight. The EO of *S*. *viticina* revealed weak porcine pancreatic lipase inhibitory activity with an IC_50_ value of 501.18 ± 0.38 µg/mL compared to that of antiobesity available drug Orlistat (IC_50_ = 12.3 ± 0.33 µg/mL). Salameh et al. reported that EO of *Micromeria fruticosa serpyllifolia* leaves (Lamiaceae) exhibited potent antilipase activity with an IC_50_ value of 39.81 µg/mL, which is better than that of *S. viticina* EO (IC_50_ = 501.18 ± 0.38 µg/mL) [18].

### 3.3. Antioxidant Activity

Due to its simplicity and high sensitivity, the widely used DPPH assay for radical scavenging was employed to assess the antioxidant activity of *S*. *viticina* EO [19]. The results of the present work showed that the EO exhibited antioxidant activity with an IC_50_ value of 19.95 ± 2.08 µg/mL, less than that of the positive control (Trolox) which has an IC_50_ value of 2.23 ± 1.57 µg/mL. Previous studies demonstrated that various *Stachys* species exhibited good antioxidant activity. A study conducted by Kukić et al. revealed that *S. anisochila*, *S. beckeana*, *S. plumose*, and *S. alpina* ssp. *dinarica* have antioxidant potential with IC_50_ values of 17.9, 20.9, 101.61, and 26.14 µg/mL, respectively [20]. The EO of *S. inflate,* growing Iran, exhibited antioxidant activity with an IC_50_ value of 89.50 ± 0.65 µg/mL [21]. Furthermore, A study performed by Conforti et al. on the antioxidant activity of the EOs extracted from various *Stachys* species from different regions of the Mediterranean area revealed that *S. palustris, S. cretica*, and *S. hydrophila* displayed the highest antiradical effect, with IC_50_ values of 48.2, 65.2 and 66.4 µg/mL, respectively [22].

### 3.4. Cyclooxygenase Inhibitory Assay

Non-steroidal anti-inflammatory molecules have the potentials to reduce inflammation and relieve pains in the human body which is associated with the increase of prostaglandin levels. However, several traditional herbal products have been used to reduce pain sensation, inflammations and fever [23]. The current cyclooxygenase inhibitory assay revealed that *S. viticina* EO caused a 98% and 99% inhibition of COX-1 and COX-2 enzymes activity at concentrations of 0.25 and 0.5 µg/mL, respectively. *S. viticina* EO exhibited cyclooxygenase inhibitory activity against COX-1 and Cox-2 with IC_50_ values of 0.25 and 0.5 µg/mL, respectively, similar to those IC_50_ values of the positive control (the NSAID etodolac).

### 3.5. Cytotoxic Activity

Cytotoxicity assays have gained increasing interest over recent years due to their great value in the biological screening of insecticide, herbicides and anticancer agents [24]. These assays are proved to be reliable, quick, cheap and reproducible. Various types of cytotoxic tests, such as luminometric, fluorometric, dye exclusion and colorimetric assays, are used in the fields of pharmacology and toxicology [25]. The significant antimicrobial activity of *S. viticina* EO encouraged us to evaluate its cytotoxicity. The potential cytotoxicity of the *S. viticina* EO towards HeLa (cervical adenocarcinoma) and Colo-205 (colon) cancer cells was evaluated using the MTS assay. As shown in Figure 1; Figure 2, treatment of Colo-205 cells with 0.5 mg/mL of *S*. *viticina* EO induced the best cytotoxic effect (90%), while treatment of HeLa cells with 2.5 mg/mL of EO induced the highest cytotoxicity (95%). The *S. viticina* EO displayed very strong cytotoxicity against HeLa and Colo-205 cancer cells.

In a previous study conducted in our laboratory, Jaradat et al. reported that *Teucrium pruinosum* EO at a dose of 7.67 mg/mL induced cytotoxic activity on HeLa cancer cells by 90–95%. The cytotoxicity of *S*. *viticina* EO against HeLa cancer cells is triple that induced by *Teucrium pruinosum* EO (Lamiaceae) [26].

### 3.6. Antimicrobial Capacity

The microdilution assay was used to assess the antimicrobial activity of *S*. *viticina* EO against MRSA, *S*. *aureus*, *E*. *faecium*, *S*. *sonnie*, *P*. *aeruginosa*, *E*. *coli*, *E*. *floccosum*, and *C*. *albicans*. The *S*. *viticina* EO inhibited the growth of all the screened pathogens. In particular, the *S*. *viticina* EO exhibited potent antibacterial activity against MRSA with MIC value of 0.039 mg/mL, which is two folds more potential than the inhibition activity against *E. coli* (MIC = 0.078 mg/mL). Moreover, the EO of *S. viticina* plant inhibited the growth of all the studied fungal strains, especially inhibited potentially the growth of *E. floccosum* with MIC value of 0.78 mg/mL.

## 4. Materials and Methods

### 4.1. Chemical Reagents

All the experiments in the current study were carried out utilizing commercially available chemicals and reagents unless otherwise stated.

### 4.2. Equipment

A gas chromatograph (Clarus 500-Perkin Elmer, Singapore), Mass Spectrometer (Clarus 560D-Perkin Elmer), microscope (IX-73-inverted, Olympus (China) CO.,LTD, Beijing, China), UV-Visible Spectrophotometer (Jenway-7315, Staffordshire, UK), microwaves-ultrasonic reactor-extractor (CW-2000, Gloria Wang Zhejiang Nade Scientific Instrument Co., Ltd., Zhejiang, China), Balance (Rad-weight, International Weighing Review, Toruńska, Poland), sonicator (MRC, 2014-207, MRC Lab., Haifa, Israel), water bath (LabTech, 2011051806, Thermo Fisher Scientific, Seoul, South Korea), incubator (Nüve, 06-3376, Ankara, Turkey), vortexer (090626691, Heidolph Company, Schwabach, Germany), autoclave (MRC, A13182, Mrc lab., Haifa, Israel), grinder (Molineux I, Jiangmen, China) and microplate reader (6000, Unilab, Fort Lauderdale, FL USA) were utilized in the current investigation.

### 4.3. Herbal Material

*Stachys viticina* leaves were collected in March 2018 from the Nablus area, Palestine. The taxonomical characterization of the plant was established at the Pharmacy Department at An-Najah National University and deposited under the voucher specimen number (Pharm-PCT-2341). Later on, *S. viticina* leaves were cleaned, rinsed at least three times with distilled water and then dried in the shade at 25 ± 2 °C and 55 ± 5 RH of humidity for three weeks. The dried leaves were cut into small pieces which were kept in special paper bags for farther experimental work [27].

### 4.4. Isolation of S. viticina Essential Oil

The EO of *S. viticina* plant was extracted using the microwave and ultrasonic apparatus as reported by Jaradat et al. [28]. Dried *S. viticina* powder (100 g) and distilled water (0.5 L) were placed in a round bottomed flask and exposed to micro- and ultrasonic- waves at a fixed power of 1000 W at 100 °C for 10 min. The extracted EO was collected into an amber glass bottle and kept in the refrigerator at 2–8 °C for further experiments.

### 4.5. GC-MS Characterization of S. viticina Essential Oil

The separation was achieved on a Perkin Elmer Elite-5-MS fused-silica capillary column (30 m × 0.25 mm, film thickness 0.25 µm)*,* using helium as carrier gas at a standard flow rate of 1.1 mL/min. The temperature of the injector was adjusted at 250 °C with an initial temperature of 50 °C, initial hold 5 min, and ramp 4.0 °C/min to 280 °C. The total running time was 62.50 min and the solvent delay was from 0 to 4.0 min. MS scan time was from 4 to 62.5 min, covering mass range 50.00 to 300.00 *m*/*z*. The chemical ingredients of the EO were characterized by comparing their mass spectra with the reference spectra in the MS Data Centre of the National Institute of Standards and Technology, and by matching their Kovats and retention indices with values reported in the literature [29,30]. In addition, the EOs Kovats and retention indices with values were compared with 20% of HPLC grade of reference EOs including *α*-pinene, eucalyptol, caryophyllene, γ-terpinene, ocimene, *endo*-borneol, *α*-terpineol, jasmone, *α*-cadinene, *p*-menthane that were purchased from Sigma-Aldrich, Hamburg, Germany) [31].

### 4.6. Porcine Pancreatic Lipase Inhibitory Assay

A porcine pancreatic lipase inhibition assay was conducted in order to assess the anti-obesity activity of *S. viticina* EO. The drug Orlistat, an anti-obesity, and anti-lipase agent, was used as a positive control. The porcine pancreatic lipase inhibitory method was performed according to the protocol described by Zheng et al. with slight modifications [32]. A 500 µg/mL stock solution from the EO was dissolved in dimethyl sulfoxide (DMSO): methanol (1:9), and five different dilutions (50, 100, 200, 300 and 400 μg/mL) were prepared. Then, a 1 mg/mL stock solution of porcine pancreatic lipase was freshly prepared before use and was dispersed in Tris-HCl buffer. The substrate used was *p*-nitrophenyl butyrate (PNPB) (Sigma-Aldrich, Hamburg, Germany), prepared by dissolving 20.9 mg of it in 2 mL of acetonitrile. 0.1 mL of porcine pancreatic lipase (1 mg/mL) and 0.2 mL of the EO from each of the concentration series were placed in 5 different working test tubes and mixed. The resulting mixture was adjusted to 1 mL by adding Tri-HCl solution and incubated at 37 °C for 15 min. thereafter, 0.1 mL of *p*-nitrophenyl butyrate solution was added to each test tube and the mixture was incubated for 30 min. at 37 °C. The pancreatic lipase activity was determined by measuring the hydrolysis of PNPB into *p*-nitrophenolate ions at 410 nm using a UV-Vis spectrophotometer. The same procedure was repeated for the positive control sample (Orlistat, Sigma-Aldrich). The inhibitory percentage of the anti-lipase activity was calculated using the following equation:Lipase inhibition% = (AB − Ats)/AB × 100%(1)
where AB is the recorded absorbance of the blank solution and Ats is the recorded absorbance of the tested sample solution.

### 4.7. α-Amylase Inhibition Assay

The *α*-amylase inhibitory activity of *S*. *viticina* EO was assessed according to the standard method reported by Nyambe-Silavwe et al. with minor modifications [33]. The EO was dissolved in DMSO (Riedel-de-Haen, Hamburg, Germany) and then diluted with a buffer ((Na_2_HPO_4_/NaH_2_PO_4_ (0.02 M), NaCl (0.006 M) at pH 6.9) to a concentration of 1000 μg/mL. A concentration series of 10, 50, 70, 100 and 500 μg/mL were prepared. 0.2 mL of porcine pancreatic *α*-amylase enzyme solution (Sigma-Aldrich, St. Louis, MO, USA) with a concentration of 2 units/mL was mixed with 0.2 mL of the EO and incubated at 30 °C for 10 min. Thereafter, 0.2 mL of freshly prepared starch solution (1%) was added and the mixture was incubated for at least 3 min. The reaction was stopped by addition of 0.2 mL dinitrosalicylic acid (DNSA) (Alf۔aAesar, Lancashire, UK), then the mixture was diluted with 5 mL of distilled water and heated in a water bath at 90 °C for 10 min. The mixture was left to cool down to room temperature and the absorbance was measured at 540 nm. A blank was prepared following the same procedure by replacing the *S*. *viticina* EO with 0.2 mL of the buffer.

Acarbose (Sigma-Aldrich, USA) was used as a positive control and prepared to adopt the same procedure described above. The *α*-amylase inhibitory activity was calculated using the following equation:% of *α*-amylase inhibition = (A_b_ – A_S_)/A_b_ 100%(2)
where A_b_ is the absorbance of the blank and A_S_ is the absorbance of the tested sample or control.

### 4.8. α-Glucosidase Inhibitory Activity Assay

The *α*-glucosidase inhibitory activity of *S*. *viticina* EO was performed according to the standard protocol with a slight modification [34]. A mixture of 50 μL of phosphate buffer (100 mM, pH 6.8), 10 μL *α*-glucosidase (1 U/mL) (Sigma-Aldrich, USA) and 20 μL of varying concentrations of *S*. *viticina* EO (100, 200, 300, 400 and 500 µg/mL) were placed in 5 different test tubes. After 15 min. incubation at 37 °C, 20 μL of pre-incubated 5 mM PNPG (Sigma-Aldrich, USA) was added as the substrate to each test tube and the reaction mixtures were incubated at 37 °C for additional 20 min. The reaction was terminated by adding 50 μL of aqueous Na_2_CO_3_ (0.1 M). The absorbance of the released *p*-nitrophenol was measured by a UV/Vis spectrophotometer at 405 nm. Acarbose at similar concentrations as the plant EO was used as the positive control. The inhibition percentage was calculated using the following equation:% of *α*-amylase inhibition = (A_b_ – A_S_)/A_b_ 100%(3)
where A_b_ is the absorbance of the blank and A_S_ is the absorbance of the tested sample or control [35].

### 4.9. Antioxidant Assay

For estimation of *S*. *viticina* EO antioxidant potential, a solution of EO (1 mg/mL) in methanol was serially diluted with methanol to obtain concentration of 1, 2, 3, 5, 7, 10, 20, 30, 40, 50, 80 and 100 µg/mL. Then, DPPH (2,2-diphenyl-1-picrylhydrazyl) reagent (Sigma, New York, NY, USA) was dissolved in 0.002% *w*/*v* methanol and mixed with the previously prepared working concentrations in a 1:1 ratio. The same procedures were repeated for Trolox (Sigma-Aldrich, Darmstadt, Denmark) which was used as a positive control. All of the solutions were kept in a dark chamber for 30 min at ordinary temperature.

Then, their absorbance values were measured at a wavelength of 517 nm utilizing a UV-Visible spectrophotometer. The DPPH inhibition potential by *S*. *viticina* EO and Trolox were determined employing the following equation:DPPH inhibition (%) = (abs_blank_ − abs_sample_)/abs_blank_ 100%(4)
where abs_blank_ is the blank absorbance and abs_sample_ is the absorbance of the samples. The antioxidant half-maximal inhibitory concentration (IC_50_) of *S*. *viticina* EO and Trolox were assessed using BioDataFit-E1051 program [36].

### 4.10. Antimicrobial Activities of S. viticina Essential Oil

The fungal and bacterial isolates used in this study were from the American Type Culture Collection (ATCC, Manassas, VA, USA) and from selected MRSA species (), that were obtained from the Palestinian area at clinical settings and exhibited multi-antibiotic resistance. The screened microorganisms included three Gram-positive bacteria (MRSA, *Enterococcus faecium* (ATCC 700221) and *S*. *aureus* (ATCC 25923)), three of Gram-negative bacteria (*Shigella sonnie* (ATCC 25931), *Pseudomonas aeruginosa* (ATCC 27853) and *Escherichia coli* (ATCC 25922) and two fungal strains (*Epidermophyton floccosum* (ATCC 10231) and *Candida albicans* (ATCC 90028)).

The microdilution assay was utilized to assess the antimicrobial activity of *S. viticina* EO against bacterial and yeast strains. A total of 8.8 g of Mueller-Hinton broth were dissolved in 400 mL of distilled water under heat and boiled for 1 min. The obtained solution was autoclaved and kept at 4 °C until use. 100 µL from the broth was placed in each well of the microdilution tray. Then, 100 µL of the EO was added to the first well and the mixture was shacked. Thereafter, serial dilutions up to well #11 were performed. Well #12 did not contain any EO and was considered as a positive control for microbial growth. A fresh bacterial colony was picked from an overnight agar culture and was prepared to match the turbidity of the 0.5 McFarland standards to provide a bacterial suspension of 1.5 × 10^8^ CFU (colony forming unit)/mL. The suspension was diluted with broth by a ratio of 1:3 to a final concentration of 5 × 10^7^ CFU. Then, 1 µL of the bacterial suspension was added to each well except well #11, which was a negative control for microbial growth. Finally, the plate was incubated at 35 °C for 18 h.

The microdilution method for the yeast *C*. *albicans* was performed as described above, except that after matching the yeast suspension with the McFarland standard, it was diluted with NaCl by a ratio of 1:50, followed by 1:20, and 100 µL was placed in the wells. The plate was incubated for 48 h instead of 18 h.

To investigate anti-*Epidermatophyton floccosum* mold activity of the *S. viticina* EO, an agar dilution method was performed. Sabouraud dextrose agar (SDA) was prepared, of which 1 mL was placed in each tube and kept in a 40 °C water bath. 1 mL of the EO was mixed with 1 mL of SDA in the first tube and serial dilutions were performed in six tubes, except tube #6, which did not contain any plant material and considered as a positive control for microbial growth.

After the SDA was solidified, the spores of the mold culture were dissolved using distilled water containing 0.05% Tween 80 and scratched from the plate for comparison with McFarland turbidity. From that suspension, 20 µL was pipetted into the six tubes, except tube #5, which was considered as a negative control. The tubes were incubated at 25 °C for 14 days. The MIC is the lowest concentration of an antimicrobial agent that inhibited the visible growth of a microorganism [37,38].

### 4.11. Determination of COX Inhibition

The *S. viticina* EO inhibitory activity of human recombinant COX-2 and ovine COX-1 enzymes was assessed using a COX inhibitor screening method (Cayman Chemical, kit No.560131, Ann Arbor, MI, USA). The yellow color of the enzymatic reaction was evaluated using UV spectrophotometer in a Microplate Reader at a wavelength of 415 nm. Cyclooxygenase inhibitory assay was performed for two concentrations of EO (0.25 and 0.5 µg/mL) with celecoxib (a commercial NSAID marker). The anti-inflammatory activity of the tested EO was determined by calculating the percentage production inhibition of prostaglandin E2 (PGE2). The tested compounds concentration causing 50% inhibition (IC_50_) of the formation of prostaglandin PGE2 by COX1 and COX2 enzymes were determined from the curve of concentration inhibition response by regression analysis [39].

### 4.12. Cell Culture and Cytotoxicity Assay

The HeLa (cervical adenocarcinoma) and Colo-205 (colon) cancer cells were cultured in media (RPMI-1640) and supplemented with 10% fetal bovine serum, 1% Penicillin/Streptomycin antibiotics and 1% l-glutamine. Cells were grown in a humidified atmosphere with 5% CO_2_ at 37 °C. Cells were seeded in 96-well plates (2.6 × 10^4^ cells). After 48 hrs, the cells were incubated with EO at various concentrations for 24 h. Cell viability was assessed by Cell-Tilter 96^®^ Aqueous One Solution Cell Proliferation (MTS) Assay following the instructions of the manufacturer (Promega Corporation, Madison, WI, USA). At the end of the treatment, 20 μL of MTS solution per 100 μL of media was added to each well and incubated at 37 °C for 2 h and finally, the absorbance was spectrophotometrically estimated at 490 nm [40].

### 4.13. Statistical Examination

All the data on *α*-amylase, *α*-glucosidase and porcine pancreatic enzymes inhibitory activity as well as on the antioxidant, COX inhibitory and cytotoxicity activities were the average of triplicate analyses. The outcomes were presented as means ± standard deviation (SD). Statistical analysis was established employing GraphPad Prism software version 6.01 (GraphPad, San Diego, CA).

## 5. Conclusions

The GC-MS analysis revealed the presence of fifty-two compounds in the *S. viticina* EO, of which *end*o-borneol was the major component, followed by eucalyptol and epizonarene. The EO of *S. viticina* showed an ability to scavenge the free radical DPPH with an IC_50_ value of 19.95 µg/mL and displayed cytotoxic activity against colon and HeLa cancer cells at 0.5 mg/mL and 1.25 mg/mL, respectively. It also showed very strong antimicrobial (against three Gram-positive, three Gram-negative bacteria and two fungi) effects, of which the strongest was against MRSA, with a MIC value of 0.039 mg/mL. Moreover, the *S. viticina* EO showed high cyclooxygenase inhibitory activity against COX-1 and COX-2, with IC_50_ values of 0.25 and 0.5 µg/mL, respectively, similar to those of the positive control (the NSAID etodolac). In addition, the EO showed potential in inhibiting the activity of *α*-amylase (56.52% at 45.22 µg/mL) and of *α*-glucosidase (69.82% at 63.09 µg/mL). Briefly, *S. viticina* EO contains pharmacologically active molecules which could be further validated and explored clinically for their therapeutic potential and for the development and design of new natural therapeutic preparations.

## Figures and Tables

**Figure 1 molecules-24-03880-f001:**
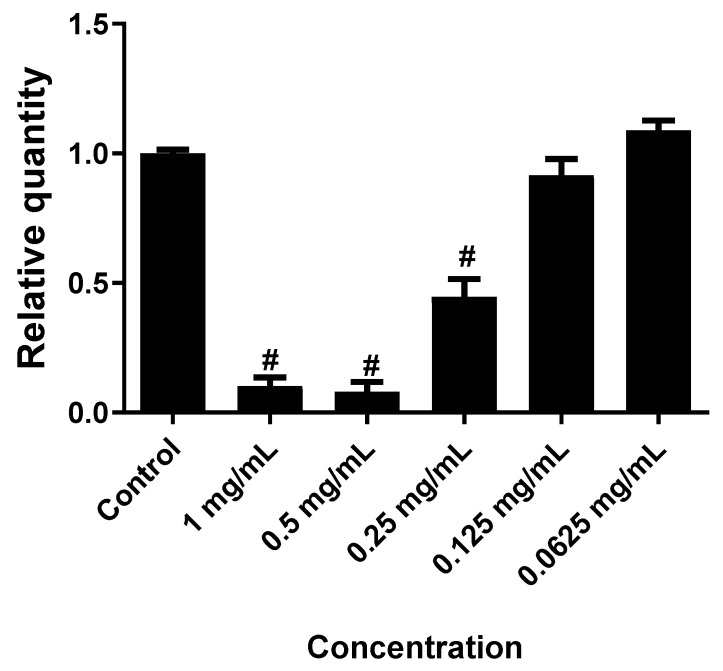
The effect of *S. viticina* EO on the cytotoxicity of Colo-205 cells. Results were depicted as relative quantities (RQs) compared to the control (only media). # where *P* < 0.0001. Error bars represent SD.

**Figure 2 molecules-24-03880-f002:**
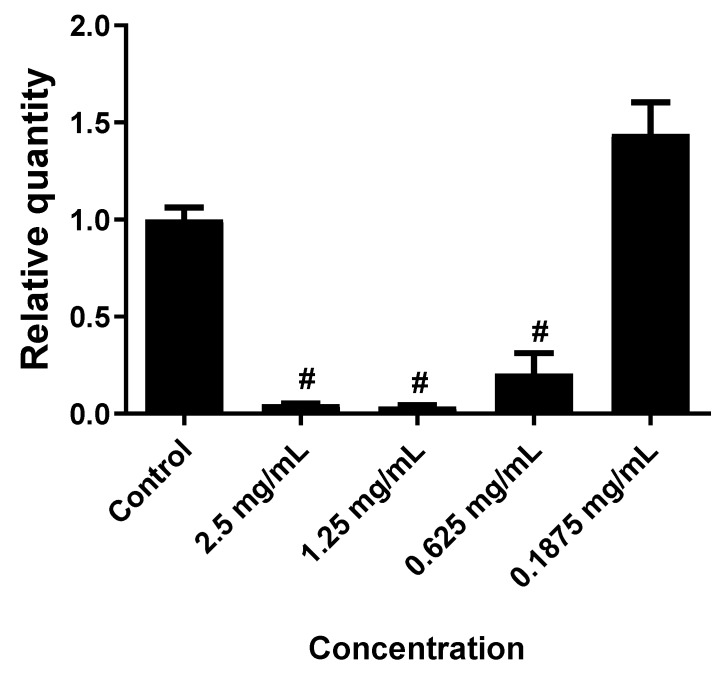
The effect of *S. viticina* EO on the cytotoxicity of HeLa cells. Results were depicted as relative quantities (RQs) compared to the control (only media). # Where *P* < 0.0001. Error bars represent SD.

**Table 1 molecules-24-03880-t001:** *S. viticina* leaves EO phytochemical composition.

	Essential Oil Components	R.T.	R.I.	%, ±SD
1.	2,4-Dimethyltetrahydro-2H-thiopyran 1,1-dioxide	5.664	770	0.11 ± 0.023
2.	α-Phellandrene	8.335	917	0.09 ± 0.001
3.	δ-3-Carene	10.271	919	0.25 ± 0.01
4.	α-Pinene	10.436	891	0.72 ± 0.013
5.	Benzylin	11.126	909	5.29 ± 0.31
6.	Santolina triene	11.361	837	0.4 ± 0.001
7.	4-Carene	12.111	893	0.69 ± 0.023
8.	Eucalyptol	12.827	841	21.26 ± 1.97
9.	γ-Terpinene	13.837	928	1.52 ± 0.014
10.	1,5-Dimethyl-1,5-cyclooctadiene	14.227	883	0.18 ± 0.001
11.	5,5-Dimethyl-2-ethyl-1,3-cyclopentadiene	14.352	881	0.18 ± 0.001
12.	Ocimene	14.778	817	0.07 ± 0.004
13.	α-Terpinolene	14.923	917	0.29 ± 0.001
14.	Linalool oxide	15.018	803	0.04 ± 0.001
15.	Arachidonic acid methyl ester	15.683	840	0.17 ± 0.02
16.	Cosmene	15.99	803	0.01 ± 0.001
17.	Norbornane	16.11	847	0.02 ± 0.001
18.	4,7-Dimethyl-4,4a,5,6-tetrahydrocyclopenta[c]pyran-1,3-dione	16.508	910	0.21 ± 0.01
19.	*cis*-Carveol	16.843	829	0.01 ± 0.001
20.	Endo-Borneol	18.584	909	29.06 ± 1.24
21.	Bornyl chloride	18.769	829	5.79 ± 0.09
22.	α-Terpineol	19.324	907	4.48 ± 0.07
23.	Menthen-9-ol	22.41	808	4.2 ± 0.023
24.	Geranyl Phenylacetate	24.966	803	0.48 ± 0.001
25.	Copaene	25.547	900	0.93 ± 0.015
26.	β-Bourbonene	25.807	921	0.78 ± 0.011
27.	Jasmone	26.137	899	0.07 ± 0.001
28.	Caryophyllene	26.967	924	0.95 ± 0.009
29.	β-Cubebene	27.302	902	0.11 ± 0.004
30.	D-Germacrene	27.753	871	0.05 ± 0.001
31.	1,1,4,8-Tetramethyl-4,7,10-cycloundecatriene	28.113	889	0.24 ± 0.006
32.	*epi*-Bicyclosesquiphellandrene	29.288	896	0.78 ± 0.008
33.	5,7-Diethyl-5,6-decadien-3-yne	29.618	929	0.06 ± 0.002
34.	γ-Cadinene	29.923	919	0.23 ± 0.008
35.	α-Cadinene	30.148	916	6.09 ± 0.1
36.	α-Ferulene	30.644	800	0.02 ± 0.001
37.	3-Ethyl-3-hydroxyandrostan-17-one	30.644	824	0.02 ± 0.002
38.	Cadala-1(10),3,8-Triene	30.779	830	0.02 ± 0.001
39.	*cis*-α-Copaene-8-ol	31.029	741	0.05 ± 0.001
40.	Bicyclo[5.2.0]nonane	31.319	818	0.03 ± 0.001
41.	Acetic Acid,4a-Methyl-2,3,4,4a,5,6,7,8 Octahydronaphthalen-2-yl	31.489	732	0.05 ± 0.001
42.	Benzoic acid, 3-hexenyl ester	31.649	843	0.07 ± 0.001
43.	Retinal	31.729	829	0.07 ± 0.001
44.	*p*-Menthane	31.864	784	0.07 ± 0.001
45.	Aromadendrane	32.649	839	0.06 ± 0.001
46.	[2,2-dimethyl-4-(3-methylbut-2-enyl)-6-methylidenecyclohexyl] methanol	32.829	875	0.02 ± 0.001
47.	α-Cubebene	33.38	844	1.48 ± 0.031
48.	Valencene	33.515	827	0.31 ± 0.002
49.	Epizonarene	33.845	855	7.89 ± 0.91
50.	γ-Gurjunene	34.45	864	2.17 ± 0.09
51.	Isophorone	34.78	880	0.53 ± 0.003
52.	Timnodonic acid	41.238	738	0.37 ± 0.001
Total				100

**Table 2 molecules-24-03880-t002:** The metabolic enzymes inhibitory and the antioxidant activities IC_50_ values of *S. viticina* essential oil compared with the required positive controls.

α-Amylase Inhibitory Activity IC_50_ (µg/mL), ±SD		α-Glucosidase Inhibitory Activity IC_50_ (µg/mL), ±SD		Antioxidant Activity IC_50_ (µg/mL), ±SD		Antilipase Activity IC_50_ (µg/mL), ±SD	
EO	Acarbose (Positive control)	EO	Acarbose (Positive control)	EO	Trolox (Positive control)	EO	Orlistat (Positive control)
45.22 ± 1.1	28.89 ± 1.22	63.09 ± 0.26	37.15 ± 0.33	19.95 ± 2.08	2.23 ± 1.57	501.18 ± 0.38	12.3 ± 0.33

**Table 3 molecules-24-03880-t003:** Antimicrobial MIC values of *S. viticina* EO.

Microorganisms	Sources	MIC, (mg/mL)
*Staphylococcus aureus*	ATCC 25923	1.5625
*Escherichia coli*	ATCC 25922	0.078
*Pseudomonas aeruginosa*	ATCC 27853	3.125
*Shigella sonnie*	ATCC 25931	3.125
Methicillin-Resistant *Staphylococcus aureus* (MRSA)	Clinical isolate	0.039
*Enterococcus faecium*	ATCC 10231	1.5625
*Candida albicans*	ATCC 90028	3.125
*Epidermophyton floccosum*	ATCC 700221	0.78

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
