# Peer review of "Fingerprinting, Antimicrobial, Antioxidant, Anticancer, Cyclooxygenase and Metabolic Enzymes Inhibitory Characteristic Evaluations of Stachys viticina Boiss. Essential Oil"

_molecules, 2019, doi:10.3390/molecules24213880_

Round 1

Reviewer 1 Report

The manuscript contains some results that have potential for publication, but still need some improvement in the presentation of the results.

The identification of the oil compounds was based on the Kovats index as indicated in the method section. But, as for the separation a temperature gradient  was used the most adequate is to use the arithmetic index according the Van den Dool and Kratz equation. Also, in the table it will be useful to present the retention indexes from the literature for each of the identified compounds. Additionally, 2,4-Dimethyltetrahydro-2H-thiopyran 1,1-dioxide does not seem to be a natural compound that will be extracted by steam distillation.

Overall the manuscript needs  justification for several of the enzymatic assays presented, there was justification for the antioxidant, antitumoral and even cyclooxygenase acivities, but not for the others. The cytotoxicity could also be presented as IC50, better than only a % of inhibition, with the respective standard deviations.

The discussion is segmented and does not reach to a final conclusion properly

Author Response

The manuscript contains some results that have potential for publication, but still need some improvement in the presentation of the results.

The identification of the oil compounds was based on the Kovats index as indicated in the method section. But, as for the separation a temperature gradient  was used the most adequate is to use the arithmetic index according the Van den Dool and Kratz equation. Also, in the table it will be useful to present the retention indexes from the literature for each of the identified compounds. Additionally, 2,4-Dimethyltetrahydro-2H-thiopyran 1,1-dioxide does not seem to be a natural compound that will be extracted by steam distillation.

Overall the manuscript needs  justification for several of the enzymatic assays presented, there was justification for the antioxidant, antitumoral and even cyclooxygenase acivities, but not for the others. The cytotoxicity could also be presented as IC50, better than only a % of inhibition, with the respective standard deviations.

Alpha-amylase, α-Glucosidase and lipase enzymes inhibitory activity assays: several experiments without   S. viticina EO were conducted, then with different concentrations of EO and compared the obtained results were compared with results of reference antiamylase and antiglucosidase drug called Acarbose, while antilipase activity was compared with antiobesity antilipase commercial drug called Orlistat. The -control without EO and +control antidiabetic and antiobesity drugs.

Regarding Cytotoxicity IC50 values: MTS is a colorimetric assay based on the reduction of the MTS tetrazolium compound by viable mammalian cells to generate a coloured formazan dye by NAD(P)H-dependent dehydrogenase enzymes in metabolically active cells. The formazan dye is quantified by measuring the absorbance at 490 nm. Then the absorbance of all samples was recorded and results were depicted as relative quantities (RQ) compared to the control (with only media; C). In other words, the absorbance of each treated sample was divided by the absorbance of the control sample (untreated cells, cells with media only). The control was normalized to 1.  Indeed, cytotoxicity can be presented in various ways including IC50 or Percent of inhibition which basically calculated by dividing the results of treated samples to the control samples.

The discussion is segmented and does not reach to a final conclusion properly

Thank you for this correction and now In the conclusion part we summarized our conducted study outcomes.

Reviewer 2 Report

Minor revisions should be made, and the manuscript should be completed and/or modified taking into account the suggestions from the attached file.

Author Response

The manuscript entitled “Fingerprinting, antimicrobial, antioxidant, anticancer, cyclooxygenase and metabolic enzymes inhibitory characters’ evaluations of Stachys viticina Boiss. volatile oil” presents the identification of chemical constituents of Stachys viticina Boiss. leaves essential oil, as well as the evaluation of its antimicrobial, anticancer, antioxidant, metabolic enzymes and cyclooxygenase (COX) inhibitory properties.

Minor revisions should be made, and the manuscript should be completed and/or modified as follows:

The numbering of the lines starts from page 8. The authors should

Done as requested

The authors are advised to use the terms essential oil instead of „volatile oil”

Done as requested

The authors should rephrase the following: „ The current study aims to recognize the chemical constituents, antimicrobial, anticancer, antioxidant, metabolic enzymes and cyclooxygenase (COX) inhibitory properties of Stachys viticina Boiss. volatile oil”

Done as requested

The authors should rephrase the following: „Furthermore, the antioxidant, cyclooxygenase and metabolic enzymes (α-amylase, α-glucosidase, and lipase) inhibitory activities were conducted using standard biomedical assays”

Done as requested

The authors should rephrase the following: „ displayed a cytotoxic, cyclooxygenase and metabolic enzymes inhibitory activities in dose depending manner”

Done as requested

The authors should rephrase the following: „ Unfortunately, these medications were and still used in the wrong ”

Done as requested

The authors should rephrase the following: „In fact, majorities of used antitumor medications are derived from natural sources and pharmacologically approved that they affect the cancer cells by various mechanisms of actions”

Done as requested

The authors should rephrase the following: „Volatile oils (VO) extracted from natural sources have been used as traditional medicine for treatment and prevention of various illnesses as well as being used widely in cosmetics, foods and food supplements in addition to their usages in dental practice and in the hygiene products”

Done as requested

The authors are advised to change the term „compromising” (page 2)

Done as requested

The authors should rephrase the paragraph „To the best of our knowledge, no previous investigations on the chemical composition of viticina VO nor on its biological and pharmacological properties have been reported. For that matter, the current investigation aims to identify the chemical constituents and to assess antioxidant, anticancer, antimicrobial, metabolic α-amylase, α-glucosidase, and lipase enzymes and cyclooxygenase (COX) inhibitory properties of S. viticina plant VO.”, since the reference

[20] presents the chemical composision of S. viticina EO. Also, they should present the vegetal product used for the study.

Done as requested

The authors are advised to use italic font for the name of species (page 3, 4, 8, )

Done as requested

The authors should better explain why they used the leaves as plant material for their study. Generally, the aerial plants of Lamiaceae species are used in

Palestinian are using S. viticina leaves only in their traditional medicine, therefore we studied only the leaves.

Also, the authors should present a comparison of their results with the ones from previously published papers on other Stachys species

Done as requested at the end of section (3.1. Phytochemical components of S. viticina  EO)

In Table 1, the authors should also present the class of each compouns (monoterpenes, sesquiterpenes), as well as the total percentage of these. Also, the average of 3 determinations ± SD for the percentage of area should be

We calculated ± SD for the percentage of area as requested

The authors drew the conclusions from lines 4-7 based on what? There is an important difference between the results presented in Table 2 for antilipase activity. The authors should better The authors should use L instead of l, for consistency (page 8, table 3)

Done

The authors should compare the results from section 2.4. with other results on Stachys species EOs

Thank you for this advice. Unfortunately, cox inhibitory activity for other Stachys species were not reported in the literature.

The authors are advised to change the terms „crude volatile oil extracts” (line 52)

Done as requested

In the Discussion section, the authors should compare their results with others, instead of presenting once again their results (subsections 3.1, 3.2, - 6)

Done as requested

The authors should rephrase the following: „Previous studies demonstrated that various

Stachys species have a wide range of antioxidant activity.” (lines 110-111)

Done as requested

Line 140 – [36]

Done as requested

The authors made only one GC-MS analysis? They should present the average of 3 determinations ± SD

Done as requested

The authors should present what EOs were used as reference from Sigma-Aldrich (linee 188)

Done as requested

The authors should rephrase the following: „A porcine pancreatic lipase inhibition assay was carried out to assess the activity of S. viticina VO in order to estimate their anti- obesity activity” (lines 190-191)

Done as requested

In the Conclusion section the authors should not repeat once again the results, they should present several conclusions related with the

Done as requested

The authors should rephrase the following: „Briefly, viticina VO is pharmacologically active molecules which can be further validated and explored clinically for its therapeutic potential and for the development and design of new natural therapeutic preparations.” (lines 324-327)

Done as requested

The authors should remove „@” from lines 317, 322,

Many thanks for your great help in improving our manuscript

Done as requested

Reviewer 3 Report

The manuscript "Fingerprinting, antimicrobial, antioxidant, anticancer, cyclooxygenase and metabolic enzymes inhibitory characters’ evaluations of Stachys viticina Boiss. volatile oil" is quite interesting and has a good scientific merit but many serious issue should be taken into consideration.

Among these points:

1- The abstract should provide the reader with many real results rather than vague sentences without numerical values. The abstract should be structured to provide more results.

2- The introduction is very long and contain many unwanted data and known facts, it will be better to rewrite to submit a more concise introduction with special emphasis on the genus itself.

3- Since the authors used the GC/MS for both identification and quantitation, the split ratio should be clearly stated because the sensitivity of the mass detector is less than the FID and the sensitivity of the used old instrument is on the border

4- The authors used the normalization method for quantitative determination of the components but they don't mention if they use a correction factor or not, the results are presented with how many replicates and also assuming that the identified components presents 100% of the components ignores the presence of unidentified compounds (quantitation should be revised)

5- Qualitative determination of the components should not based on comparison with NIST or Wiley databases only, A better method to use only NIST because the version of the used databases is not written. (It is highly recommended to include a column in table one to cite the reported KI in comparison with the calculated KI), In addition since the Area %  is used no need for the exact area 

6- It is widely accepted that borneol and endo-borneol are synonymous, if the authors using them as different isomers it should be corrected.

7- The GCMS chromatogram and most of the figures (2-4) should be shifted to the supplementary since it is not suitable to present the same results with a table and different figures.

8- The authors should confirm that the unit used is correct since a common mistake is to use microgram as if it is synonymous with microliter  in volatile oil research but they completely different so they must mentioned and confirm the used units.

9- Since the table should represent a standalone part in the manuscript, the full name of the used microorganisms should be written (the ATCC number should be given )

10- A positive control for the cytotoxic study is also required 

11- The Latin name of the plant and all other plants in the references should be italics

Author Response

Comments and Suggestions for Authors

The manuscript "Fingerprinting, antimicrobial, antioxidant, anticancer, cyclooxygenase and metabolic enzymes inhibitory characters’ evaluations of StachysviticinaBoiss. volatile oil" is quite interesting and has a good scientific merit but many serious issue should be taken into consideration.

Among these points:

1- The abstract should provide the reader with many real results rather than vague sentences without numerical values. The abstract should be structured to provide more results.

We agree with the respected reviewer and we rewrote all the abstract to provide more results.

2- The introduction is very long and contain many unwanted data and known facts, it will be better to rewrite to submit a more concise introduction with special emphasis on the genus itself.

Thank you and now we concise introduction as requested.

3- Since the authors used the GC/MS for both identification and quantitation, the split ratio should be clearly stated because the sensitivity of the mass detector is less than the FID and the sensitivity of the used old instrument is on the border

Split ratio is the column carrier gas flow rate divided by the split vent flow rate. These information depends on the procedure used and our apparatus autmaticly calculated the RT and RT.

4- The authors used the normalization method for quantitative determination of the components but they don't mention if they use a correction factor or not, the results are presented with how many replicates and also assuming that the identified components presents 100% of the components ignores the presence of unidentified compounds (quantitation should be revised)

We think that almost all the components of the S. viticina EO were identified as revealed from the GC-MS chromatogram.

5- Qualitative determination of the components should not based on comparison with NIST or Wiley databases only, A better method to use only NIST because the version of the used databases is not written. (It is highly recommended to include a column in table one to cite the reported KI in comparison with the calculated KI), In addition since the Area %  is used no need for the exact area 

Regarding the NIST database and 20% of the used reference EOs were identified S. viticina EOs and we did not add the reported KI because the computer accordingly identified the components of the EO. We deleted the column of area as requested and added the used reference compounds.

6- It is widely accepted that borneol and endo-borneol are synonymous, if the authors using them as different isomers it should be corrected.

We agree with respected reviewers and accordingly to his advices we corrected Table 1.

7- The GCMS chromatogram and most of the figures (2-4) should be shifted to the supplementary since it is not suitable to present the same results with a table and different figures.

The GCMS chromatogram and most of the figures (2-4) were moved to the supplementary.

8- The authors should confirm that the unit used is correct since a common mistake is to use microgram as if it is synonymous with microliter  in volatile oil research but they completely different so they must mentioned and confirm the used units.

Done

9- Since the table should represent a standalone part in the manuscript, the full name of the used microorganisms should be written (the ATCC number should be given )

Done

10- A positive control for the cytotoxic study is also required 

MTS is a colorimetric assay based on the reduction of the MTS tetrazolium compound by viable mammalian cells to generate a coloured formazan dye by NAD(P)H-dependent dehydrogenase enzymes in metabolically active cells. The formazan dye is quantified by measuring the absorbance at 490 nm. Then the absorbance of all samples is recorded and results were depicted as relative quantities (RQ) compared to the control (with only media; C). In other words, the absorbance of each treated sample was divided by the absorbance of the control sample (untreated cells, cells with media only). The control was normalized to 1.

11- The Latin name of the plant and all other plants in the references should be italics

Done

Many thanks for your helpful advices

Round 2

Reviewer 3 Report

The authors have responded to most of the raised points however a positive control results should be added otherwise the impact of biological work will be very limited

Author Response

Thank you for your kind advice which really improved our manuscript and all the required positive controls were added as requested.

Thank you